# Dietary Factors and Modulation of Bacteria Strains of *Akkermansia muciniphila* and *Faecalibacterium prausnitzii*: A Systematic Review

**DOI:** 10.3390/nu11071565

**Published:** 2019-07-11

**Authors:** Sanne Verhoog, Petek Eylul Taneri, Zayne M. Roa Díaz, Pedro Marques-Vidal, John P. Troup, Lia Bally, Oscar H. Franco, Marija Glisic, Taulant Muka

**Affiliations:** 1Institute of Social and Preventive Medicine, University of Bern, 3012 Bern, Switzerland; 2Corlu Cancer Early Diognosis and Training Center, 59100 Tekirdag, Turkey; 3Department of Medicine, Internal Medicine, Lausanne University Hospital (CHUV), 1011 Lausanne, Switzerland; 4Standard Process Inc Nutrition Innovation Center, Kannapolis, NC 28018, USA; 5Department of Diabetes, Endocrinology, Clinical Nutrition and Metabolism, Bern University Hospital, 3010 Bern, Switzerland; 6Leibniz Institute for Prevention Research and Epidemiology-BIPS, 28359 Bremen, Germany

**Keywords:** *Akkermansia muciniphila*, *Faecalibacterium prausnitzii*, microbiome, systematic review, randomized controlled trials, dietary interventions

## Abstract

*Akkermansia muciniphila* and *Faecalibacterium prausnitzii* are highly abundant human gut microbes in healthy individuals, and reduced levels are associated with inflammation and alterations of metabolic processes involved in the development of type 2 diabetes. Dietary factors can influence the abundance of *A. muciniphila* and *F. prausnitzii*, but the evidence is not clear. We systematically searched PubMed and Embase to identify clinical trials investigating any dietary intervention in relation to *A. muciniphila* and *F. prausnitzii*. Overall, 29 unique trials were included, of which five examined *A. muciniphila,* 19 examined *F. prausnitzii*, and six examined both, in a total of 1444 participants. A caloric restriction diet and supplementation with pomegranate extract, resveratrol, polydextrose, yeast fermentate, sodium butyrate, and inulin increased the abundance of *A. muciniphila*, while a diet low in fermentable oligosaccharides, disaccharides, monosaccharides, and polyols decreased the abundance of *A. muciniphila*. For *F. prausnitzii*, the main studied intervention was prebiotics (e.g. fructo-oligosaccharides, inulin type fructans, raffinose); seven studies reported an increase after prebiotic intervention, while two studies reported a decrease, and four studies reported no difference. Current evidence suggests that some dietary factors may influence the abundance of *A. muciniphila* and *F. prausnitzii.* However, more research is needed to support these microflora strains as targets of microbiome shifts with dietary intervention and their use as medical nutrition therapy in prevention and management of chronic disease.

## 1. Introduction

The gut microbiota, defined as the complex, diverse, and vast microbial community resided in the human gut, is emerging as a key player in the pathophysiology of several chronic conditions [1]. Gut microbiota extract energy from nutrients and regulate several biological processes. A microbiota imbalance, such as low abundance of *Akkermansia muciniphila* and *Faecalibacterium prausnitzii,* favours inflammatory processes, potentially leading to inflammatory disorders of the gastrointestinal tract such as irritable bowel syndrome and inflammatory bowel disease, as well as to colorectal cancer [2,3,4,5,6]. Also, increases in *A. muciniphila* and *F. prausnitzii* abundance can regulate metabolic functions and appear to exert protective effects against the development of obesity [7], type 2 diabetes (T2D) [8], and atherosclerosis [9]. Therefore, these two bacteria have been considered as potential bioindicators of human cardiometabolic health and conditions where underlying inflammation plays a role. *A. muciniphila* is a mucin-degrading bacterium of the phylum Verrucomicrobia, while *F. prausnitzii* is an important butyrate-producer of the phylum of Firmicutes. *F. prausnitzii* is also the major bacterium of the *Clostridium leptum* group [10,11]. It accounts for 5% of total bacteria in faeces and it is an important source of energy for the colonocytes [12,13].

Modulating the abundance of *A. muciniphila* and *F. prausnitzii* in the intestinal flora may contribute to prevent and treat inflammatory and cardiometabolic diseases [14,15,16]. Studies have shown that dietary factors can influence the diversity and composition of the gut microbiome more than human genetic factors [17]. Data from animal studies suggest that polyphenols-rich diets and specific phytochemicals such as curcumin and epigallocatechin gallate may improve the abundance of *A. muciniphila* and *F. prausnitzii* in the gut microbiome [18,19,20,21,22]. A clinical trial with obese insulin-resistant patients showed that resveratrol, a natural phenol, increased the abundance of *A. muciniphila* in Caucasians but not in other ethnic groups [23]. A diet high in fermentable oligo-, di- and monosaccharides and polyphenols (FODMAP), which has been implicated in digestive diseases such as irritable bowel syndrome, inflammatory bowel disease, and Crohn’s disease, resulted in changes in gut microbiota including changes in *A. muciniphila* and *F. prausnitzii* but the evidence is not consistent [24,25]. Furthermore, intake of dietary fibres, such as the soluble fibre inulin, has been associated with increased abundance of *A. muciniphila* and *F. prausnitzii* in healthy and overweight/obese individuals, but another study showed no influence of inulin intake on *F. prausnitzii* [26,27,28]. Despite the emerging evidence of the impact of dietary factors on changing the microbiome composition, a comprehensive assessment of the dietary interventions in modulating *A. muciniphila* and *F. prausnitzii* is lacking. Identifying dietary microbiome modulators may help understanding how diet and phytochemicals can be employed in medical nutrition therapy to increase the abundance of *A. muciniphila* and *F. prausnitzii*.

Therefore, the aim of this systematic review is to identify, integrate and discuss all available evidence from clinical trials in humans on the association of diet with changes in *A. muciniphila* and *F. prausnitzii*. 

## 2. Material and Methods

### 2.1. Literature Search

This review was conducted and reported in accordance with the PRISMA [29] and MOOSE [30] guidelines (Appendix A). PubMed and Embase were used to identify published studies from database inception until April 23, 2019 (date last searched) that examined the associations between any dietary factor and changes in *A. muciniphila* and *F. prausnitzii* in humans. To capture all relevant studies, we used a broad search strategy using terms related to *A. muciniphila* and *F. prausnitzii* (Appendix B). We did not apply any restrictions with regard to language and date. Furthermore, to identify additional relevant studies, we checked the reference lists of the studies included in the current review.

### 2.2. Study Selection Criteria

To be included in the review, studies had to full-fill all of the following criteria: (i) were clinical trials (randomized placebo-controlled trials, randomized trials comparing different exposures and single-arm trials); (ii) included subjects 18+ years, and (iii) reported associations between any dietary intervention and absolute and relative abundance of *A. muciniphila*, or *F. prausnitzii*, or both. As we were interested to understand the effect of dietary factors on gut microbiome, we decided to include all trials irrespective of their aim; therefore, all eligible trials that reported baseline and end of study information on outcomes of interest were included in this review. Studies conducted in animals, children/teenagers, conference abstracts, letters to the editor, interventions examining probiotics solely and non-interventional studies were excluded. Following the selection criteria, two reviewers independently screened titles and abstracts for eligibility. Next, two reviewers assessed the full-text of potentially eligible studies. In cases of disagreement, a decision was made by consensus or, if necessary, a third reviewer was consulted.

### 2.3. Data Extraction

Two reviewers extracted the data independently using a predesigned form including study design, study population, location, age range, dietary intervention and control (if applicable), duration of intervention and study results.

### 2.4. Assessing the Quality of Trials

Two reviewers evaluated the risk of bias within each study using “The Cochrane Collaboration’s tool” [31]. Studies were judged to be at low or high risk of bias based on criteria to evaluate random sequence generation, allocation concealment, blinding of participants/personnel and outcome assessment, incomplete outcome data and selective reporting. Trials were considered to be in low risk of bias if allocation concealment, blinding of participants and outcome assessors were all coded “yes”, if a compliance assessment was done and the number of dropouts and reasons for dropout were reported; otherwise, the trials were considered to be at high risk of bias. If the risk of bias could not be determined in any of the segments, (e.g., information not provided) the risk of bias was classified as unknown. Lastly, for each trial, an overall score of quality was provided. Good quality was assigned when all criteria were met (i.e. low risk of bias for each domain). Fair quality was assigned when one criterion was not met (i.e. high risk of bias for one domain) or two criteria were unclear, and the assessment that this was unlikely to have biased the outcome, and there is no known important limitation that could invalidate the results. Poor quality was assigned when (i) one criterion was not met or two criteria were unclear, and the assessment that this was likely to have biased the outcome, and there are important limitations that could invalidate the results, or (ii) when two or more criteria were listed as high or unclear risk of bias.

## 3. Results

### 3.1. Study Identification and Selection

In total, we identified 1433 relevant citations. After screening based on titles and abstracts, 59 citations were selected for detailed full text evaluation. Of those, 30 articles [23,24,25,26,27,28,32,33,34,35,36,37,38,39,40,41,42,43,44,45,46,47,48,49,50,51,52,53,54,55], based on 29 unique trials, met the selection criteria and were included in the review (Figure 1). Among these, 19 articles examined *F. prausnitzii* [26,27,32,34,36,37,38,39,41,42,44,46,47,49,50,51,52,54,55], five articles examined *A. muciniphila* [23,28,35,43,48], and six articles examined both [24,25,33,40,45,53].

### 3.2. Characteristics of the Included Trials

Of the 29 unique included trials, 15 were conducted in Europe, 8 in North America, 4 in Australia and New Zealand, 1 in Iran, and 1 in China; 22 were conducted among both men and women, 4 among men and 3 among women. Overall, the trials reported results for 1444 participants. All trials were published after 2005 and 80 percent in the last 5 years. The baseline age of the participants was 18+. The duration of the interventions ranged from 2 weeks to 6 months. The design of the included trials comprised randomized double-blind placebo-controlled trials, single-arm trials, multi-arm parallel trials, cross-over trials, and single and multi-centre trials. Detailed characteristics of the included trials are presented in Table 1 and Table 2. Two trials scored an overall good quality score, indicating a low risk of bias in all domains (Appendix A), whereas for the other trials the overall score was fair (*n* = 10) or poor (*n* = 18). There was a low risk of bias for all trials in the domain sequence generation and allocation concealment. For five trials, the risk of bias in the domain blinding of participants and personnel was unclear. In the domain blinding of assessment, 16 trials had an unclear risk of bias, as this was not clearly stated in these trials. Nine trials did not report how they handled missing data, and were therefore classified as being unclear in the assessment of risk of bias in the domain of missing data. High risk of bias was found in the domain of selective reporting for 14 trials, as they did not report effect estimates. The domain “other bias” included the risk of false positive findings for 14 trials, as these trials did not account for multiple testing in their analysis.

### 3.3. Trials Examining A. muciniphila

Eleven trials examined the association between a dietary intervention and *A. muciniphila* [23,24,25,28,33,35,40,43,45,48,53]. The dietary interventions examined were a caloric restriction diet, a reduced energy diet, a diet low in fermentable oligo-, di-, mono-saccharides and polyols (FODMAP), supplemental fibres, a yeast fermentate (EpiCor), sodium butyrare and inulin, pomegranate extract, kiwifruit capsules, and resveratrol. Findings are summarized in Table 1.

Dao M et al. evaluated a caloric restriction diet compared to a weight stabilization diet among overweight and obese participants [35]. In the caloric restricted group, the abundance of *A. muciniphila* decreased in participants with a high baseline level of *A. muciniphila* and increased in participants with low *A. muciniphila* levels. In the weight stabilization diet group, the abundance of *A. muciniphila* decreased in participants with both low and high baseline levels of *A. muciniphila*. Similarly to Dao M et al. [13], Medina-Vera I et al. examined a reduced energy diet compared to placebo in patients with T2D [45]. They reported that consumption of the reduced-energy diet increased levels of *A. muciniphila* by approximately 125%. However, two other studies coming from the same cross-over trial examined a diet low in FODMAPs compared to a typical Australian diet containing FODMAPs, and showed that the typical Australian diet increased absolute and relative abundance of *A. muciniphila* [24,25].

A cross-over trial conducted showed that polydextrose supplementation increased *A. muciniphila* compared to soluble corn fibre supplementation or placebo (i.e. no supplemental fibre control) among healthy individuals [40]. Another cross-over trial reported no changes in relative or absolute abundance of *A. muciniphila* after increasing the intake of resistant starch and wheat bran foods among healthy individuals and patients with ulcerative colitis in remission [53]. Supplementation with EpiCor, a yeast fermentate, increased *A. muciniphila* compared to placebo (maltodextrin) in healthy individuals with moderate symptoms of gastrointestinal discomfort [48]. Supplementation with sodium butyrate or inulin increased *A. muciniphila* compared to placebo (starch) in overweight and obese diabetic patients, but no significant differences were found with combined sodium butyrate and inulin supplementation [28].

Pomegranate extract increased *A. muciniphila* by 47-fold in healthy metabolite urolithin A producers compared to non-producers [43]. However, kiwifruit capsules did not have a significant effect on *A. muciniphila* abundance in healthy and functionally constipated individuals [33]. Resveratrol supplementation led to an increase in *A. muciniphila* in USA Caucasians, but not in other ethnic groups [23].

### 3.4. Trials Examining F. prausnitzii

Twenty-five studies examined the effect of dietary interventions on *F. prausnitzii* [24,25,26,27,32,33,34,36,37,38,39,40,41,42,44,45,46,47,49,50,51,52,53,54,55]. Of these 25 studies, 17 studies examined prebiotics and the other studies examined isoflavones, dietary fat, carbohydrates, a ketogenic diet, kiwifruit capsules, sun-dried raisins, iron therapy, and a Chinese herbal formula. The findings are summarized in Table 2.

Treatment with prebiotic inulin-type fructans led to an increase of *F. prausnitzii* compared to placebo (maltodextrin) in obese women [36]. Supplementation of fructo-oligosaccharides, which are prebiotic fructans, increased the level of *F. prausnitzii* compared to placebo (maltodextrin) in patients with diarrhoea or mixed irritated bowel syndrome [41]. Supplementation of prebiotic inulin-oligofructose also increased the level of *F. prausnitzii* in healthy individuals [26]. Moreno-Indias I. et al. studied the prebiotic effect of red wine in male metabolic syndrome patients and healthy individuals and reported an increase of *F. prausnitzii* for red wine intake compared to baseline [46]. Another study found that *F. prausnitzii* was greater when healthy men consumed polydextrose or soluble corn fiber supplementation, which could be potential prebiotics, than when they consumed no supplemental fiber [40]. Medina-Vera I. et al. examined the effect of a reduced energy diet with prebiotic properties compared to a placebo diet in patients with type 2 diabetes, and reported an increase of *F. prausnitzii* of 34% [45]. Fernando WMU et al. studied the prebiotic potential of chickpea oligosaccharides (raffinose) alone or as components of chickpea and found that *F. prausnitzii* was more abundant in the raffinose diet and the chickpea diet compared to the control diet [38]. Ingestion of butyrylated high amylose maize starch led to greater relative increases of *F. prausnitzii* compared to ingestion of low amylose maize starch [55].

However, there are also studies reporting a decrease in *F. prausnitzii* after a prebiotic intervention. Patients from the intensive care unit starting nasogastric enteral nutrition receiving additional oligofructose/inulin had significantly lower concentration of *F. prausnitzii* compared to patients receiving placebo [44]. A combination of epigallocatechin-3-gallate and resveratrol decreased *F. prausnitzii* compared to placebo in obese men but not in obese women [47]. A cross-over trial comparing a fibre-free enteral formula or a formula supplemented with dietary fibre consisting of pea fibre and fructo-oligosaccharides showed large reductions in the number of *F. prausnitzii* during both the fibre-free and fibre-supplemented diets among healthy participants [52].

Moreover, some studies reported no differences in *F. prausnitzii* after a prebiotic intervention. Halmos E. et al. examined a diet low in FODMAPs compared to a diet containing FODMAPs in two different studies in patients with irritated bowel syndrome and patients with clinically quiescent Crohn’s disease [24,25]. They did not report significant differences in *F. prausnitzii* between the diets in both studies. Another study conducted among men with Crohn’s disease examining the effect of fructo-oligosaccharides compared to a non-prebiotic carbohydrate also did not found differences in *F. prausnitzii* [32]. Vulevic J. et al. investigated the effect of administering Bi2muno (B-GOS) compared to placebo (maltodextrin) in a cross-over trial among overweight subjects predisposed to the development of metabolic syndrome, and reported no significant effects on counts of the *F. prausnitzii* cluster during the study [54]. Increasing the intake of resistant starch and wheat bran foods containing fibres did not change the relative and absolute abundance of *F. prausnitzii* in healthy participants and patients with ulcerative colitis in remission [53]. Among healthy individuals, Ramnani P. et al. studied the prebiotic effect of Jerusalem artichoke compared to a placebo, but did not report significant differences in *F. prausnitzii* [27]. 

Clavel T. et al. studied the effect of a prebiotic (isoflavones and fructo-oligosaccharides) or probiotic (isoflavones and *B. animalis* DN-173 010) compared to placebo (isoflavones alone) in postmenopausal women [34]. They reported that percentages of *F. prausnitzii* decreased significantly in control subjects compared to probiotic and prebiotic groups. Guadamuro L. et al. examined the effect of isoflavone concentrate supplementation alone in menopausal women and found an increase in the intensity of *F. prausnitzii* after isoflavone supplementation [39].

Fava F. et al. showed that the type and quantity of dietary fat and carbohydrate can alter faecal microbiome in individuals at increased risk of metabolic syndrome, whereby *F. prausnitzii* increased after intervention of a diet with high monounsaturated fatty acids and a high glycaemic index, compared to diets with high saturated fatty acids, high monounsaturated fatty acids in combination with a low glycaemic index, or with high carbohydrates in combination with high or low glycaemic index [37]. A ketogenic diet including a minimum of 0.8-1 gram per kilogram of bodyweight of protein from animal sources did not have an effect on *F. prausnitzii* [49]. Consumption of kiwifruit capsules increased *F. prausnitzii* abundance in functionally constipated individuals, compared to placebo [33]. Addition of sun-dried raisins to the diet also increased the abundance of *F. prausnitzii* among healthy individuals [50]. Oral iron therapy resulted in lower abundance of *F. prausnitzii* compared to intravenous iron therapy among iron deficient inflammatory bowel disease patients [42]. Xu J et al. examined the effect of a Chinese herbal formula (Gegen Qinlian Decoction) in a low, medium and high dose in recently diagnosed type 2 diabetes patients. They reported that all three doses of Gegen Qinlian Decoction treatment significantly enriched *F. prausnitzii* compared with baseline [51].

## 4. Discussion

In the current systematic review of clinical trials, diet-induced changes in *A. muciniphila* and *F. prausnitzii* were reviewed. An increase in abundance of *A. muciniphila* was observed after a caloric restriction diet, supplementation with pomegranate extract, resveratrol, polydextrose, EpiCor or sodium butyrate, whereas a diet low in FODMAPs decreased the abundance of *A. muciniphila*. Prebiotics use was the main dietary intervention investigated in relation to *F. prausnitzii*, showing contradictory results depending on the type of prebiotics. Isoflavone supplementation and intake of some types of fatty acids, such as monounsaturated fatty acids, was associated with increased abundance of *F. prausnitzii*, whereas inulin intake and a reduced energy diet increased the abundance of both *A. muciniphila* and *F. prausnitzii*.

Approximately 100 trillion microorganisms are present in the body but mostly residing in the gastrointestinal tract [56]. In response to environmental changes, the gut microbiome can affect gene expression in humans. While around 23,000 genes make up the human genome, the microbiome consists over 3 million genes producing thousands of metabolites, and therefore affecting many aspects of human health, including gastrointestinal and cardiometabolic phenotypes [56,57]. Changes in microbiota are observed in patients with inflammatory gastrointestinal disease, prediabetes, and T2D, particularly a decreased abundance of *A. muciniphila* and *F. prausnitzii* [58]. In addition, low abundance of *A. muciniphila* and *F. prausnitzii* has been associated with increased inflammatory processes and atherosclerosis, and *F. prausnitzii* transplantation has been shown to be an effective therapeutic approach for diabetes and its complications [59]. *F. prausnitzii* is an important butyrate-producer with anti-inflammatory properties, while *A. muciniphila* degrades mucin in the gut lining resulting in syntrophic interactions and stimulation of intestinal metabolite pool [60]. For example, co-cultivations of *A. muciniphila* with butyrate-producing bacteria resulted in syntrophic growth and butyrate production [60]. Butyrate plays an important role in anti-inflammation and maintaining intestinal barrier integrity by modulating intestinal macrophages’ function, downregulating lipopolysaccharide-induced pro-inflammatory mediators, such as nitric oxide, IL-6, and IL-12, inducing the differentiation of regulatory T cells and stabilizing hypoxia-inducible factor [61,62,63,64]. Further, butyrate can have beneficial effects on glucose and energy homeostasis by activating intestinal gluconeogenesis, as well as inducing apoptosis of colon cancer cells [65]. Studies in animal models have shown that colonization by *A. muciniphila* resulted in transcriptional changes, leading to an increase in the expression of genes associated with immune responses and cellular lipid metabolism [11,66]. Besides degrading mucins, *A. muciniphila* can also stimulate mucin production, suggesting an autocatalytic process [67,68]. Mucins are large, highly glycosylated proteins that play a crucial role in luminal protection of the gastrointestinal tract, thereby reducing translocation of pro-inflammatory lipopolysaccharides, controlling fat storage, adipose tissue metabolism, and glucose homeostasis [68,69]. Therefore, altering the abundance and dynamics of these two strains of bacteria can lead to alterations in metabolic processes, and prevention and management of metabolic diseases.

Diet is the main source of energy to humans, but can also modulate microbiota and impact host-microbe interactions [70]. Therefore, gut microbiota may be crucial in mediating the health effects of foods. The gut microbiota provides the exoenzymes to catalyze and ferment substrates that cannot be completely digested by the human digestive tract to produce metabolites such as short fatty acids including butyrate [71]. Such non-digestible substrates include complex carbohydrates derived from plants and dietary fibres from cereals, legumes, vegetables, fruits, and nuts. Modulation of the gut microbiome through adherence to a high-fibre plant-based diet has been suggested as a potential therapeutic approach for prevention and treatment of inflammatory gastrointestinal diseases and metabolic diseases such as T2D and restoration of microbiota function [72,73]. In line with this hypothesis, we found in the current review that inulin-type fructans, fructo-oligosaccharides, polydextrose or soluble corn fiber supplementation, and raffinose can lead to increase abundance of *A. muciniphila* and/or *F. prausnitzii*. However, we found no consistent associations or decreases for other types of prebiotics and abundance of *F. prausnitzii* such as inulin-oligofructose and resistant starch or wheat bran foods. Yet, this inconsistency is somehow expected considering that studies used different types of prebiotics, for which bacteria have differing specificity and therefore affecting differently the number of fermentable substrates in the gut [14]. In addition, different types of fibres have differing effects in the luminal pH and transit rate. Type of fibre being consumed, gut transit time, and the functional capabilities of gut microbiota influences the fermentation of carbohydrate and subsequent regional delivery of metabolites [14]. Hence, the different results we see in this study on the association between prebiotics and *F. prausnitzii* may reflect this complexity. This is also in line with the findings from observational studies showing that some types of dietary fibres (such as those from cereals) but not all are associated with reduced risk of T2D [74].

Pomegranate extract, a rich and varied source of polyphenolic compounds, and resveratrol polyphenol have been widely investigated for their antioxidant, anti-inflammatory, anti-diabetic and anti-atherogenic properties in both genders [75,76]. Phytoestrogens seem to have similar effects but mainly in women; use of phytoestrogens in women is associated with improvement of menopausal symptoms, better glycaemic control and reduced risk of T2D [77,78]. However, the mechanisms of action of these compounds remain unclear. Microbiome plays an important role in metabolizing the non-absorbed fraction of these compounds, and therefore defining the ability of these compounds such as polyphenols to exert health effects [79]. On the other hand, pomegranate and polyphenol resveratrol can modulate the abundance of *A. muciniphila*, while isoflavone phytoestrogens alone, and not in combination with fibres, can modulate the abundance of *F. prausnitzii* [19,23,34,39,43]. While it is not completely clear how such compounds can affect intestinal microorganisms, limited evidence suggests that it could be either by the selective pressure they exert on specific microorganisms, or by modifying environmental conditions of the intestinal tract [34].

Modifying the diet can lead to rapid changes in microbiota within the first days. A study reported human gut microbiome changes after only 24 hours after shifting between plant and animal protein-based diets [80]. Similarly, the studies included in this review showed that dietary interventions are associated with changes in *A. muciniphila and F. prausnitzii* in the first weeks of intervention [26,41,44,50]. This suggests that diet may be an important and fast modifier of *A. muciniphila and F. prausnitzii,* and the microbiome in general, and a promising future therapeutic approach to prevent and treat inflammatory and metabolic conditions associated with specific microbiome strains. Our study highlights several additional factors that might be important to consider for a more comprehensive understanding of diet-related diseases and to elucidate the links between nutrition and dynamics of *A. muciniphila* and *F. prausnitzii*. However, dietary effects on microbiome may depend on health status, the baseline abundance of bacteria strains already in the gut, ethnicity, medication, and sex [24,25,32,33,34,35,37,41,42,45,46,51]. This could also explain the inconsistent results across studies in this systematic review, as we included studies among different sexes, ethnicities and health statuses. For instance, men and women have different genetic background, energy and nutritional requirements across the lifespan, as well as differences in gastrointestinal transit time, which can contribute to sex differences in microbiome [81,82]. Therefore, considering the sex-differences in the gut microbiome can provide novel insights in tailoring interventions and treatment, and therefore improving precision medicine. Machine-learning algorithms can help to predict inflammatory metabolic responses to meals, microbiome and how that differs by sex and ethnicity, and thus help to improve the emerging concept of personalized nutrition. For example, Zeevi et al. [83] used a machine-learning algorithm to predict personalised glucose responses after meals based on clinical and gut microbiome data. Based on the algorithm, a dietary intervention was designed and successfully shown to normalize blood glucose levels in a double-blinded randomised crossover trial of 26 patients [83]. Future research based on a large population-based studies and clinical trials is needed to evaluate a greater variety of food components and establish whether such personalised nutritional approaches based on microbiome composition are feasible and can improve clinical outcomes.

Further, it may be important to understand the differences that whole foods and ultra-processed foods may have on gut microbiota and subsequently on health outcomes. It has been shown that fibre-rich ultra-processed foods change the microbiota towards a less diverse and less beneficial composition compared to fibre-rich whole foods [84]. Processed foods have been undergone high heat and pressure, which can lead to several chemical and physical changes, including inactivation of endogenous enzymes, increased content of soluble dietary fibre and mechanical damage to the cell walls [85]. This suggests that whole unprocessed foods will enter the colon in an intact state which could favour the growth of bacteria that degrade fibre and produce beneficial metabolites, and thus exerting protective effects on health outcomes [85]. This theory is also supported from epidemiological studies showing that whole foods such as whole fish and whole grains can have a more beneficial effect than processed fish and refined grains, and that processed meat is in general associated with a stronger adverse health effect than whole meat [86]. Therefore, investigating the health effects of nutrients present in ultra-processed and whole food in the context of the microbiome can shed light on the mechanisms of the adverse effects of the Western diet.

To our knowledge, this is the first systematic review on the subject that critically and systematically appraised the literature on the effect of dietary interventions on *A. muciniphila* and *F. prausnitzii.* Nevertheless, some limitations from the included studies in this review merit careful consideration. Firstly, not all studies reported baseline and/or end of trial estimates between groups, and therefore the interpretation and the strength of the reported associations remained unclear. This and other factors (such as different types of study populations, interventions, and study designs) also limited providing a meaningful quantitative pooling of the existing data. It is important that future studies report summary estimates for all bacteria strains under investigation, and not apply selective reporting. Further, the studies included in this review were of suboptimal quality and the conclusions should be interpreted with caution. Most studies have been published in recent years and were conducted in limited sample sizes (16 studies had ≤30 participants) within interventions lasting short time periods (15 studies lasted for ≤6 weeks). Moreover, we cannot exclude the possibility that the changes we observed in *A. muciniphila* and *F. prausnitzii* are due to changes that dietary factors can lead to in the health condition of the participants, rather than the effect of diet alone. Nevertheless, most of clinical trials included in this review were performed in healthy individuals, and therefore we would expect that it is less likely that dietary factors we examined alter the abundances of the bacteria strains through changes in health. We also cannot exclude the possibility that the dietary factors lead to changes in other bacterial strains, which might have an effect on *A. muciniphila* and *F. prausnitzii*. What appears to be a promising field for intervention to improve health is also highly complex and new studies are required to untangle these complex interactions of diet, microbiome, and diseases in larger populations.

## 5. Conclusions

In summary, we found that specific dietary interventions could influence the abundance of *A. muciniphila* and *F. prausnitzii*, highlighting a novel potential mechanism describing how diet can affect inflammatory and metabolic health outcomes. Understanding the links between different types of dietary interventions and changes in abundance of *A. muciniphila* and *F. prausnitzii* can help shaping future nutritional recommendations and designing medical nutrition therapies that can help prevent and treat diseases related to these two strains of bacteria such as inflammatory gastrointestinal disease and T2D.

## Figures and Tables

**Figure 1 nutrients-11-01565-f001:**
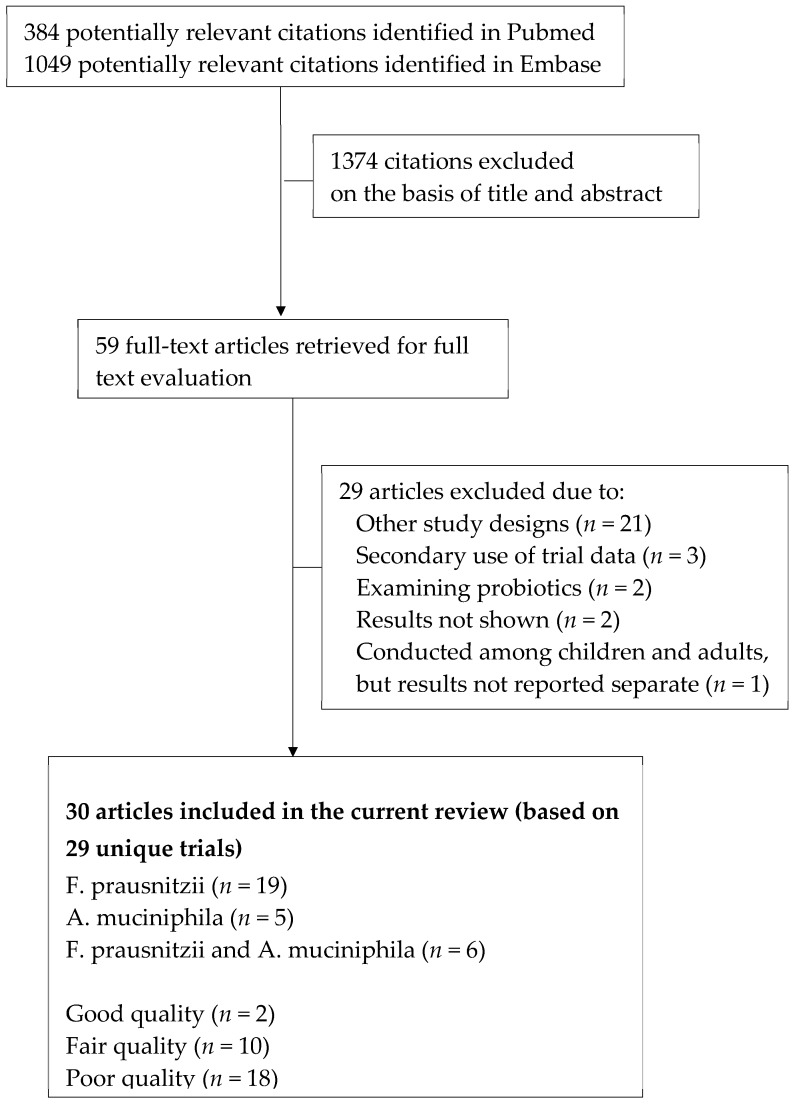
Flowchart of studies included in the current review.

**Table 1 nutrients-11-01565-t001:** Descriptive Summary of Randomized Clinical Trials Investigating the Associations Between Dietary Interventions and Akkermansia Muciniphila.

Lead Author, Publication Year	Study Design	Location/Age Range	Individual Health Status	Total Participants	Sex	Period of Intervention	Dietary Treatment Characteristics		Main Findings
							Intervention Type	Control Type	
Blatchford P et al. 2017* [1]	Randomized double-blind placebo-controlled cross-over trial	New Zealand/23–56	Healthy participants with no clinical symptoms of constipation and functionally constipated participants	29	W and M	4 weeks each intervention (2 weeks washout period between each intervention)	ACTAZIN™ (600 mg/d) green kiwifruit extract low dose	Placebo (isomalt coloured green 2400 mg/d)	*A. muciniphila* was significantly more abundant in the functionally constipated group, but no effect of the interventions on *A. muciniphila.*
							ACTAZIN™ (2400 mg/d) green kiwifruit extract high dose		
							Livaux™ (2400 mg/d) gold kiwifruit extract		
Dao M et al. 2015 [13]	Single-arm cross-over trial	France/41.9 ± 12.3	Overweight and obese participants	49	W and M	12 weeks	Caloric restriction diet (1200 kcal/d for W and 1500 kcal/d for M)	Weight stabilization diet (prescribed individually by a dietitian)	Caloric restriction diet: Subjects with *A. muciniphila* at or above the median had a decrease in abundance of *A. muciniphila* while in the group with *A. muciniphila* lower than median there was an increase. The difference was statistical significant. Weight stabilization diet: In both subjects with *A. muciniphila* at or above and lower, there was a decrease in abundance of *A. muciniphila* with no difference between the groups.
Halmos EP et al. 2015* [16]	Single-arm blinded randomized cross-over trial	Australia/18+	Healthy participants and participants with irritable bowel syndrome	33	W and M	6 weeks	Diet low in FODMAPs	Diet containing FODMAP content of a typical Australian diet	Typical Australian diet increased absolute and relative abundance for mucus-associated *A. muciniphila* (*p* < 0.001).
Halmos EP et al. 2016* [17]	Single-arm blinded randomized cross-over trial	Australia/18+	Patients with clinically quiescent Crohn’s disease	9	W and M	6 weeks	Diet low in FODMAPs	Diet containing FODMAP content of a typical Australian diet	Relative abundance was higher for mucus-associated *A. muciniphila* during the Australian compared with low FODMAP diet (*p* = 0.016).
Hooda S et al. 2012* [6]	Randomized double-blind placebo-controlled cross-over trial	USA/27.5 ± 4.33	Healthy participants	25	M	9 weeks	Polydextrose (PDX) (7 g, 3 times per day)	Placebo (no supplemental fiber control (NFC) (0 g, 3 times per day))	*A. muciniphila* was greater after PDX intake than after the NFC or SCF treatment (*p* < 0.05).
							Soluble corn fiber (SCF) (7 g, 3 times per day)		
James SL et al. 2015* [54]	Randomized single-blind cross-over trial	Australia/18–72	Patients with UC in remission and healthy subjects	29	W and M	8 weeks	‘Low resistant starch (RS)/wheat bran (WB)’ foods containing 2–5 g RS and 2–5 g WB fibre per day	NA	Patients with UC had a lower abundance of *A. muciniphila*. For both cohorts, increasing the intake of RS/WB gave no indication of changes in relative or absolute abundance.
							‘High RS/WB’ foods containing 15 g RS and 12 g WB fibre per day		
Li Z et al. 2015 [20]	Single-arm trial	USA/28.9 ± 8	Healthy volunteers	20	W and M	4 weeks	Pomegranate extract (1000 mg)	NA	The data were not shown for the overall population. *A. muciniphila* was 33 (at baseline) and 47 fold (after 4 weeks) higher in stool samples of Urolithin A producers compared to non-producers.
Medina-Vera I et al. 2019* [9]	Randomized, double-blind placebo-controlled trial	Mexico/30–60	Patients with Type 2 Diabetes	81	W and M	3 months	A reduced-energy diet with a dietary portfolio (DP) (14 g of dehydrated nopal, 4 g of chia seeds, 30 g of soy protein and 4 g of inulin)	Placebo (28 g of calcium caseinate and 15 g of maltodextrin)	DP consumption increased levels of *A. muciniphila* by approximately 125%.
Pinheiro I et al. 2017 [11]	Randomized, double-blind placebo-controlled trial	Belgium/20–69	Healthy with reduced bowel movements and other symptoms of GI discomfort stratified in severe and moderate	80	W and M	6 weeks	EpiCor fermentate (500 mg/d)	Placebo (maltodextrin (500 mg/d))	Significant relative increase of *A. muciniphila* in the moderate GI discomfort symptoms group at visit week 3 (*p* = 0.0001) and visit week 6 (*p* = 0.036).
Roshanravan N et al. 2017 [25]	Randomized, double-blind placebo-controlled trial	Iran/30–55	Overweight and obese diabetes patients	60	W and M	6 weeks	Group A: Butyrate group (600 mg/d sodium butyrate + inulin placebo)	Butyrate + inulin placebo (6 starch capsules (100 mg) and 10 g of starch powder)	The percentage changes of *A. muciniphila* abundance indicated a significant increase in group taking sodium butyrate and inulin (group A and B) in comparison with the placebo group (*p* < 0.05). A non-significant rise in this bacterium concentration was seen after supplementation with both sodium butyrate and inulin (group C).
							Group B: inulin group (10 g/d inulin powder + butyrate placebo)		
							Group C: butyrate + inulin group (600 mg/d sodium butyrate + 10 g/d inulin powder)		
Walker JM et al. 2019 [27]	Randomized, double-blind placebo-controlled trial	USA/30–70	Obese insulin resistant subjects with metabolic syndrome	28^a^	M	5 weeks	Resveratrol (500 mg Mega-RES 99% capsules twice daily)	Placebo (two 500 mg placebo capsules twice daily)	Overall, there was no difference. However, when split by ethnicity, resveratrol administration to Caucasian subjects led to an increase in *A. muciniphila* compared to the non-Caucasians.

* Studies examining both A. Muciniphila and F. Prausnitzii. FODMAPs Fermentable Oligosaccharides, Disaccharides, Monosaccharides and Polyols; FOS fructo-oligosaccharides; GI gastrointestinal; M men; NA not applicable; UC ulcerative colitis; W women. ^a^ Stool samples collected in 16 subjects.

**Table 2 nutrients-11-01565-t002:** Descriptive Summary of Randomized Clinical Trials Investigating the Associations Between Dietary Interventions and Faecalibacterium prausnitzii.

Lead Author, Publication Year	Study Design	Location/Age Range	Individual Health Status	Total Participants	Sex	Period of Intervention	Dietary Treatment Characteristics		Main Findings
							Intervention Type	Control Type	
Benjamin JL et al. 2011 [12]	Randomized double-blinded placebo-controlled trial	UK/39.5 ± 14.4	Patients with Crohn’s disease	103	M	4 weeks	Normal diet supplemented with 15 g/day FOS, comprising fructose polymers of differing chain lengths	Placebo (maltodextrin 15 g/day)	No significant differences between patients in the FOS and placebo group at week 4 (*p* = 0.95).
Benus RFJ et al. 2010 [52]	Randomized double-blinded cross-over trial	UK/21–34	Healthy	14	W and M	4 weeks	A formula supplemented with dietary fibre (14 g/l) consisting of pea fibre and fructo-oligosaccharides	NA	There were large and statistically significant reductions in the numbers of the *F. prausnitzii* group during both the fibre-free and fibre-supplemented diets. No differences between the fibre-free and fibre-supplemented diet (*p* = 0.23).
							A fibre-free enteral formula		
Blatchford P et al. 2017* [1]	Randomized double-blind placebo-controlled cross-over trial	New Zealand/23–56	Healthy participants who had no clinical symptoms of constipation and functionally constipated participants	29/W and M	W and M	4 weeks each intervention (2 weeks washout period between each intervention)	ACTAZIN™ L (600 mg/d)	Placebo (isomalt coloured green 2400 mg/d)	*F. prausnitzii* abundance significantly increased from 3.4 to 7.0% following Livaux™ supplementation in the functionally constipated group (*p* = 0.024).
							ACTAZIN™ H (2400 mg/d)		
							Livaux™ (2400 mg/d)		
Clavel T et al. 2005 [2]	Randomized double-blind placebo-controlled trial	France/60.4 ± 7.1	Postmenopausal women	39	W	30 days	Probiotic group: isoflavones (100 mg/d) + *B. animalis* DN-173 010	Placebo (isoflavones 100 mg/d)	Bacterial percentages for *F. prausnitzii* subgroup decreased significantly in control subjects compared to the probiotic and prebiotic group (*p* = 0.034).
							Prebiotic group: isoflavones (100 mg/d) + FOS (7 g/d)		
Dewulf EM et al. 2012 [3]	Double-blind placebo-controlled trial	Belgium/ 47.5 ± 8.5	Obese	30	W	3 months	ITF prebiotics (Synergy 1, namely, inulin/oligofructose 50/50 mix)	Placebo (maltodextrin)	Treatment with ITF prebiotics, but not the placebo, led to an increase in *F. prausnitzii.*
Fava F et al. 2013 [14]	Five-arm parallel, placebo-controlled, single-blind study	UK/56.0 ± 9.5	Individuals at increased risk of metabolic syndrome	88	W and M	24 weeks	High SFA diet	NA	Numbers of *F. prausnitzii* increased after intervention with high CHO and low GI (*p* = 0.022) and high SFA (*p* = 0.018) diet compared to baseline.
							High MUFA/high GI		
							High MUFA/Low GI		
							High CHO/High GI		
							High CHO/Low GI		
Fernando WMU et al. 2010 [4]	Randomized cross-over trial	Canada/25.6 ± 8.7	Healthy	12	W and M	9 weeks	Control diet + 5 g/d raffinose	Control diet	*F. prausnitzii* was more abundant in the raffinose diet and the chickpea diet compared to the control diet.
							Control diet + 200 g/d canned chickpea		
Guadamuro L et al. 2015 [15]	Single-arm trial	Spain/48–61	Menopausal women with no chronic disease	16	W	24 weeks	One tablet isoflavoneconcentrate (80 mg) per day	NA	There was an increase in the intensity of *F. prausnitzii*.
Halmos EP et al. 2015* [16]	Single-arm blinded randomized cross-over trial	Australia/18+	Irritable bowel syndrome and healthy individuals	33	W and M	6 weeks	Diet low in FODMAPs	Diet containing FODMAP content of a typical Australian diet	Low FODMAP diet reduced total bacterial abundance, but did not impact relative abundance of *F. prausnitzii.*
Halmos EP et al. 2016* [17]	Single-arm blinded randomized cross-over trial	Australia/18+	Patients with clinically quiescent Crohn’s disease	9	W and M	6 weeks	Diet low in FODMAPs	Diet containing FODMAP content of a typical Australian diet	No significant difference in *F. prausnitzii* between the two diets.
Hooda S et al. 2012* [6]	Randomized double-blind placebo-controlled cross-over trial	USA/27.5 ± 4.33	Healthy	25	M	9 weeks	Polydextrose (PDX) (7 g, 3 times per day)	Placebo: no supplemental fiber control (NFC) (0 g, 3 times per day)	*F. prausnitzii* was greater in participants when they consumed PDX or SCF than when they consumed NFC (*p* < 0.05).
							Soluble corn fiber (SCF) (7 g, 3 times per day)		
Hustoft TN et al. 2016 [7]	Randomized double-blind placebo-controlled cross-over trial	Norway/18–52	Diarrhea-predominant or mixed irritated bowel syndrome	20	W and M	10 days each intervention (3 weeks washout period)	Fructo-oligosaccharides (FOS) 16 g/d	Placebo: Maltodextrin 16 g/d	Ten days of FOS supplementation increased the level of *F. prausnitzii*.
James SL et al. 2015* [54]	Randomized single-blind cross-over trial	Australia/18–72	Patients with UC in remission and healthy subjects	29	W and M	8 weeks	‘Low resistant starch (RS)/wheat bran (WB)’ foods containing 2–5 g RS and 2–5 g WB fibre per day	NA	For both cohorts, increasing the intake of RS/WB gave no indication of changes in relative or absolute abundance in *F. prausnitzii*.
							‘High RS/WB’ foods containing 15 g RS and 12 g WB fibre per day		
Lee T et al. 2017 [19]	Randomized, double-blind placebo-controlled trial	Canada/18+	Iron deficient Inflammatory bowel disease patients	72	W and M	12 weeks	Oral iron sulfate 300 mg, tablet, twice a day	Iron sucrose, 300 mg, intravenous, three or four/day	Lower abundance of *F. prausnitzii* after oral iron therapy compared to intravenous iron therapy (*p* = 0.009).
Majid HA et al. 2014 [8]	Multi-centre, randomized double-blind controlled trial	UK/70.8 ± 9.7	Patients from the ICU starting exclusive nasogastric enteral nutrition	22	W and M	Up to 14 days	Oligofructose/inulin 7 g/d	Placebo: maltodextrin 7 g/d	There were significantly lower concentrations of *F. prausnitzii* in patients receiving additional oligofructose/inulin (*p* = 0.01).
Medina-Vera I et al. 2019* [9]	Single-centre randomized, controlled, double-blind parallel-group trial	Mexico/30–60	Patients with Type 2 Diabetes	81	W and M	3 months	A reduced-energy diet with a dietary portfolio (DP) comprising 14 g of dehydrated nopal, 4 g of chia seeds, 30 g of soy protein and 4 g of inulin	Placebo, comprising of 28 g of calcium caseinate and 15 g of maltodextrin.	Dietary intervention with functional foods significantly modified faecal microbiota compared with placebo. DP consumption for 12 weeks increased levels of *F. prausnitzii* by approximately 34%.
Moreno-Indias I et al. 2016 [21]	Randomized, cross-over controlled trial	Spain/45–50	Metabolic syndrome and healthy individuals	20	M	10 weeks (75 days)	Red wine, 272 mL/day	De-alcoholized (no ethanol) red wine, 272 mL/dat	In metabolic syndrome patients, there was a significant increase of *F. prausnitzii*, after the red wine and de-alcoholized red wine intake periods compared to baseline. In the healthy group, a significant increase in the number of *F. prausnitzii* through the intervention period was observed.
Most J et al. 2017 [10]	Randomized double-blind placebo-controlled trial	The Netherlands/20–50	Obese	42	W and M	12 weeks	A combination of epigallocatechin-3-gallate (EGCG) and resveratrol (RES) supplements (EGCG + RES; 282 and 80 mg/day, respectively)	Placebo (partly hydrolyzed microcrystalline cellulose-filled supplements)	EGCG+RES supplementation significantly decreased Bacteroidetes and tended to reduce *F. prausnitzii* in men (*p* = 0.05 and *p* = 0.10, respectively) but not in women (*P* = 0.15 and *P* = 0.77, respectively).
Ramirez-Farias C et al. 2008 [23]	Randomized, cross-over trial	UK/38.1 ± 2.43	Healthy adults	12	W and M	3 weeks	Inulin–oligofructose, 5 g, twice daily	Did not consume any supplement	*F. prausnitzii* exhibited a significant increase after intervention (*p* = 0.019).
Ramnani P et al. 2010 [24]	Three-arm parallel, placebo-controlled, double-blind study	UK/18–50	Healthy adults	60	W and M	3 weeks intervention (3 weeks washout period)	Jerusalem artichoke (JA) inulin- predominantly made of pear-carrot-sea buckthorn and JA juices or purées (PCS); two 100 mL shots per day	Placebo: Water-based preparation with added sugar, thickened and flavoured with blood orange, carrot and raspberry extracts and flavours (but no juice or purees)	No significant differences during the intervention and washout period.
							JA inulin- predominantly made of plum-pear-beetroot and JA juices or purées (PPB); Two 100 mL shots per day		
Tagliabue A et al. 2017 [26]	Single-arm trial	USA/18–34	Glucose Transporter 1 Deficiency Disorder (GLUT1-DS) patients	6	W and M	12 weeks	Ketogenic diet including a minimum of 0.8–1 gram per kilogram of body weight of protein from animal sources (e.g., eggs, milk, meat, poultry and fish)	NA	There was no statistical significant difference.
Vulevic J et al. 2013 [55]	Randomized double-blind placebo-controlled cross-over trial	UK/45.2 ± 11.9	Overweight subjects predisposed to the development of metabolic syndrome	45	W and M	12 weeks each intervention (4 week washout period)	Bi^2^muno (B-GOS)	Placebo (maltodextrin)	The two dietary interventions had no significant effects on counts of total bacteria and *F. prausnitzii* cluster during the study.
West NP et al. 2013 [56]	Randomized double-blind controlled trial	Australia/37.4 ± 8.4	Healthy active cyclists	41	W and M	28 days	Ingestion of 40 g/day of butyrylated high amylose maize starch (HAMSB)	Low amylose maize starch (LAMS)	There were relative greater increases in faecal *F. prausnitzii* (5.1-fold; *p* < 0.01) in the HAMSB group.
Wijayabahu AT et al. 2019 [28]	Single-arm trial	USA/18–59	Healthy individuals	13	W and M	2 weeks	Sun-dried raisins: Three servings per day; one serving contained 28.3 g raisins and 2 grams of dietary fiber	NA	*F. prausnitzii* significantly increased after the first week of raisin intake and this increase continued during the second week of raisin consumption (*p* < 0.05).
Xu J et al. 2015 [29]	Randomized, double-blind placebo-controlled clinical trial	China/8.5 ± 2.6	Recently diagnosed type-2 diabetes patients	187	W and M	12 weeks	Low dose of Gegen Qinlian Decoction	Placebo	All three doses of GQD treatment significantly enriched *F. prausnitzii* compared with baseline.
							Medium dose of Gegen Qinlian Decoction		
							High dose of Gegen Qinlian Decoction		

* Studies examining both A. Muciniphila and F. Prausnitzii. CHO Carbohydrate; GI Glycemic index; FODMAPs Fermentable Oligosaccharides, Disaccharides, Monosaccharides and Polyols; FOS fructo-oligosaccharides; ICU intensive care unit; ITF Inulin-type fructans; M men; MUFA monosaturated fat; SFA Saturated fatty acids; UC ulcerative colitis; W women.

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
