# Peer review of "Dietary Factors and Modulation of Bacteria Strains of Akkermansia muciniphila and Faecalibacterium prausnitzii: A Systematic Review"

_nutrients, 2019, doi:10.3390/nu11071565_

Reviewer 1 Report

General comment

This study is well designed and the criteria used for selection of literatures are also convincing. However, I have several comments to be answered by the authors.

Specific comments

1. This review properly describes the current progresses of studies on A. muciniphilaand F. Prausnitzii. However, several studies showed conflict results. The authors need to discuss possible reasons clearer.

2. In the present manuscript, the effects of food factors on the distribution of A. muciniphilaand F. Prausnitziiwere reviewed without considering if the influences were derived as the cause or result of changes in the health condition of subjects. The authors need to discuss them respectively. I mean, some food improve health condition by increase of the strains while others increase the strains accompanying with changes of health condition.

3. Related to the above comment, the authors need to describe the change of health conditions accompanying with the change of the distribution ofA. muciniphilaand F. Prausnitzii more detail.

4. I would like to know how the food described in this review specifically affect the distribution of A. muciniphilaand F. Prausnitzii. Do you find any relation between the distribution of them and some other specific strains?

5. I would like to know the reason why the authors exclude the studies regarding children. Are the effects of food factors reviewed in this manuscript on the distribution of A. muciniphilaand F. Prausnitziidifferent between adults and children?

6. Minor point. ‘and’ on line 331 should be removed

Reviewer 2 Report

Akkermansia and Faecalibacterium are two major bacterial genera that have been implicated in diseases. It is worthwhile to summarize the clinical trials on modulating these two genera, although all the 29 clinical trials by far were based on small sample size with short follow-up time. Most had low quality.

Comments:

Please incorporate quality level in Figure 1, the last frame.

Summarize bias in a more systematic way related to quality level.

In tables, the summary for study findings was not always complete.   Any comparison made with baseline in intervention groups? It not sure whether at the baseline levels, the intervention groups and placebo groups had the same microbial profile

The author may want to cluster the summary based on the trial design, such as cross-over, or single arm. etc.

In tables 1 and 2 or a separate table, please describe what sequencing method was used in each study. If 16S rRNA sequencing was used, please state which region was targeted (such as V3,V4…). Did all studies use feces? How was it collected and stored?

Tables, Total participants/sex column is confusing.

Please insert table row heading in separate pages.

Line 243, were reviewed, not observed.

Author Response

Round  2

Reviewer 1 Report

The authors replied to comments by this reviewer well.